

# An Unmanned Aerial Vehicle Sampling Platform for Atmospheric Water Vapor Isotopes in Polar Environments

Kevin S. Rozmiarek[1], Bruce H. Vaughn[1], Tyler R. Jones[1], Valerie Morris[1], William B. Skorski[1], Abigail G. Hughes[1], Jack Elston[2], Sonja Wahl[3], Anne-Katrine Faber[3], Hans Christian Steen-Larsen[3]

[1] Institute of Arctic and Alpine Research, University of Colorado, Boulder, CO, USA
[2] Black Swift Technologies, Boulder, CO 80301, USA
[2] Geophysical Institute, University of Bergen and Bjerknes Centre for Climate Research, Bergen, 5020, Norway

*Correspondence to*: Kevin S. Rozmiarek (kevin.rozmiarek@colorado.edu)

**Abstract.** Above polar ice sheets, atmospheric water vapor exchange occurs across the planetary boundary layer (PBL) and

is an important mechanism in a number of processes that affect the surface mass balance of the ice sheets. Yet, this exchange is not well understood, and has substantial implications for modeling and remote sensing of the polar hydrologic cycle. Efforts to characterize the exchange face substantial logistical challenges including the remoteness of ice sheet field camps, extreme weather conditions, low humidity and temperature that limits the effectiveness of instruments, and dangers associated with flying manned aircraft at low altitudes. Here, we present an Unmanned Aerial Vehicle (UAV) sampling

platform for operation in extreme polar environments that is capable of sampling atmospheric water vapor for subsequent measurement of water isotopes. This system was deployed to the East Greenland Ice-core Project (EastGRIP) camp in northeast Greenland during summer 2019. Six sampling flight missions were completed. With a suite of atmospheric measurements onboard the UAV (temperature, humidity, pressure, GPS) we determine the height of the PBL using on-line algorithms, allowing for strategic decision making by the pilot to sample water isotopes above and below the PBL. Water

isotope data was measured by a Picarro 2130-i instrument using flasks of atmospheric air collected within the nose cone of the UAV. The internal repeatability for δD and δ$^{18}$O was 2.8 ‰ and 0.45 ‰, respectively, which we also compared to independent EastGRIP tower-isotope data. Based on these results, we demonstrate the efficacy of this new UAV-isotope platform and present improvements to be utilized in future polar field campaigns. The system is also designed to be readily adaptable to other fields of study, such as measurement of carbon cycle gases or remote sensing of ground conditions.

## 1 Introduction

The Greenland and Antarctic ice sheets interact with the atmosphere through continuous exchange of water vapor by condensation and sublimation, and through precipitation events (*Fettweis et al. 2019*). The planetary boundary layer (PBL, the lowest layer of the troposphere directly influenced by the surface) generally has a thickness of 10s to 100s of meters above the ice sheet, and exchanges water vapor with the free troposphere (FT) (*Helmig et al.* 2002, *Galewsky et al.* 2016). It

is not clear how much water vapor is exchanged from surface sublimation flux, nor if the exchange ultimately results in a



significant mass loss or mass gain for the ice sheet (*Boisvert et al.*, 2017). The exchange of water vapor between the ice sheet and different parts of the atmosphere has importance for varying fields of study, including 1) ice-atmosphere modeling and mixing processes, 2) ice sheet mass balance, 3) satellite detection algorithms, 4) moisture tracking, 5) ice core science and 6) modeling of the hydrologic cycle in general. In each of these cases, a critical missing component is the reliable measurement

of the water vapor flux across the PBL border with the free troposphere. We hypothesize that atmospheric water vapor isotopes at altitudes within or above the PBL and especially in the few hundred meters above the ice sheet represents a measurable quantity, which allows us to quantify this flux.

Stable isotopes have been used to characterize the hydrological cycle since the first precipitation observations made by

*Dansgaard* (1954). More recent studies have treated transport, phase changes, and other factors not available from precipitation records alone (e.g. *Galewsky et al.* 2016). For ice sheets, a common assumption that has persisted since early studies is that the isotopic composition of the ice sheets is solely informed by precipitation events. Yet, this assumption is being overturned with clear evidence that the ice sheet and the atmosphere constantly exchange water isotopologues with different rates leading to post-depositional change in the snow isotopic composition (*Steen-Larsen et al.* 2013, 2014; *Ritter et*

*al.* 2016; *Hughes et al.* 2021). This paradigm shift has not been fully accounted for in models, nor are these findings utilized for constraining ice sheet-atmosphere interactions. This forms a substantial motivation for this study.

Early attempts to measure atmospheric water vapor isotopes were made by cryogenically trapping water vapor for subsequent analysis of the liquid, typically with mass spectrometers (*Arnason, 1969*) and over the ice sheet (*Steen-Larsen et*

*al.* 2011, *Landais et al.* 2012). With the advent of laser based isotopic instruments (*Baer et al.* 2002, *Crosson et al.* 2002, *Gupta et al.* 2009, *Iannone, et al.* 2010), measurements in remote locations have become much more feasible, including the polar-regions (*Steen-Larsen et al.* 2013, *Bastrikov et al.* 2014, *Bonne et al.* 2019, *Leroy-Dos et al.* 2020). A comprehensive listing of atmospheric water vapor isotopic measurements can be found in *Wei, Z. et al.* (2019). Direct measurements of water vapor isotopes collected from various elevations on small towers above the ice surface in Greenland (*Steen-Larsen et*

*al.*, 2013, *Berkelhammer et al.* 2016, *Madsen et al.* 2019) along with laboratory experiments (*Ebner et al.* 2017) have opened the pathway to understanding vapor transport and exchange with surface snow. In addition, satellite measurements (e.g. *Worden et al.* 2006, *Frankenberg et al.* 2009) and ground-based remote sensing data using spectra measured within global networks (*Schneider et al.* 2012, 2017; *Rokotyan et al.* 2014) offer greatly increased spatial coverage and typically measure the total atmospheric column. However, because of the different vertical sensitivities of $H_2^{16}O$, $H_2^{18}O$ and HDO of columnar

retrievals, these data must be used carefully. So far, modeling based on water stable isotope observations of the exchange between the PBL and free troposphere has only been done for the marine boundary layer and only using ground-based observations (*Benetti et al.* 2018).



Bridging the two different scales of satellite remote sensing and *in situ* ground-based measurements is a challenging necessity for understanding the hydrologic cycle. Most efforts and testing have occurred at lower-latitudes, far from the ice sheet. *Franz and Röckmann* (2005) developed a cryogenic sampler and protocol to collect stratospheric water vapor, from very small mixing ratios (<10 ppm) flown on a C-17 aircraft during flights between New Zealand and Antarctica. In 2007, *Strong et al.* was successful in using pre-evacuated 650 cc glass flasks to collect atmospheric water vapor samples in the

field, then cryogenically extracting the water and reducing it to hydrogen (*Friedman et al.* 1953), followed by mass spectrometer analysis. Vertical profiles were collected in approximately 300 m intervals using a light manned-aircraft with a ceiling of 2–3 km above ground level (AGL) in the desert southwest of the U.S. (*Strong et al.* 2007). As the engine of the aircraft was turned off during sampling in the *Strong el al.* study, obtaining airborne samples near the surface would be too dangerous.


There have been two recent measurement campaigns that utilized *in-situ* optical water vapor isotope instruments to constrain remote-sensing water isotope products. *Herman et al.* (2014) utilized a Picarro L1115-i CRDS analyzer across 27 flights by a Navion L-17a aircraft in the lower and mid troposphere over the Alaskan boreal forest in a bias estimation study with the remote Aura Tropospheric Emission Spectrometer (TES). They estimated up to a +37‰ $\delta D$ bias in the TES PBL estimate

with a 20‰ uncertainty in that bias. *Dryoff et al.* 2015 flew seven profiles of $\delta D$ with a CASA C-212 aircraft with onboard ISOWAT-II instrument over the Canary Islands to triangulate between ground based Fourier transform infrared (FTIR) spectrometer measurement and space-based IASI (infrared atmospheric sounding interferometer) during the MUSICA campaign (MUlti-platform remote Sensing of Isotopologues for investigating the Cycle of Atmospheric water). A validation study estimated a 40‰ uncertainty of $\delta D$ in the lower troposphere and 15‰ in the upper troposphere against the FTIR

product. Uncertainty in IASI was estimated by *Schneider et al.* in 2015 to be 15‰ in the mid troposphere with a +30-70‰ bias. Uncertainties of this magnitude are inadequate for constraining water vapor across the PBL and remain a target for improved methodologies.

We present results from a UAV pilot study at the East Greenland Ice-core Project (EastGRIP) site in northeast Greenland,

occurring in summer 2019. We describe how customized UAVs can now be used to safely bridge satellite and ground-based measurements, all while overcoming the challenging polar conditions to sample atmospheric air in the low-to-mid troposphere above the Greenland Ice Sheet. This is accomplished by designing an effective yet relatively inexpensive sampling platform with 3D-printed parts and accessible control devices on a commercially available fixed wing UAV that collects air samples aloft for analysis immediately following flight with ground-based instrumentation. We show that water

vapor isotope measurements can be achieved with sufficient precision relative to the magnitude of the observed gradient across the PBL, and comparable with independent measurements made at the EastGRIP 10m tower. We also demonstrate that algorithmic methods of evaluating clustering indices of real-time on board sensors to determine the altitude of the PBL, which can be used by the flight team to make informed sampling decisions mid-flight. We make recommendations for future





field deployments to polar ice sheets and discuss the potential for how the observations can be used to improve the scientific
understanding of varying fields of study.

## 2 Methods

### 2.1 Water Isotope Measurements

In this study, we made atmospheric water vapor measurements at the EastGRIP ice core field site in northeast Greenland
(75.63°N, 35.99°W; 2,700 m above sea level). A cavity ring-down laser spectroscopy (CRDS) instrument, model L2130-*i*
(Picarro Inc., Santa Clara, CA) was used in conjunction with a custom inlet to introduce both samples and standards with
equal treatment, described in more detail in Section 2.6. The standard water isotope data was analyzed on a continuous flow
analysis (CFA) system adapted from *Jones et al.* (2017a). Results were validated against measurements made by the
SNOWISO project (H2020 European Research Council Start Grant #759526), also using a Picarro L2130-*i* instrument
(Section 2.3).

The data consist of measurements of hydrogen and oxygen isotopes in water vapor, where the ratio of heavy to light water
isotopes in a sample is expressed in δ notation (*Epstein et al.* 1953, *Mook* 2000) relative to Vienna Standard Mean Ocean
Water (VSMOW) and normalized to Standard Light Antarctic Precipitation:

$$\delta_{sample} = \left[ \left( \frac{R_{sample}}{R_{VSMOW}} \right) - 1 \right] * 1000 \qquad (1)$$

where $R$ is the isotopic ratio $^{18}O/^{16}O$ or D/H (i.e., $^{2}H/^{1}H$). The δD and δ$^{18}$O symbols refer to fractional deviations from
VSMOW, normally expressed in parts per thousand (per mille or ‰).

### 2.2 EastGRIP Hydrological Cycle

The hydrological cycle on the Greenland ice sheet has several isotopic reservoirs and exchanges (Figure 1). The dominant
reservoir is the ice sheet, composed of ice, firn and snow with a relatively positive water isotope value compared to the
overlying atmosphere (*Steen-Larsen et al.* 2011). At the ice sheet/atmosphere interface, both radiative (shortwave and
longwave) and non-radiative (sensible and latent heat) energy fluxes occur, affecting the energy mass balance of the ice
sheet. The summation of these processes leaves a diurnal imprint on the water isotopes in the upper few centimeters of the
firn (*Ritter et al.* 2016, *Madsen et al.* 2019, *Hughes et al.* 2021). Within the PBL, turbulent mixing occurs with a magnitude
largely dependent on stratification and wind shear. Significantly stable stratification of the PBL (e.g. during polar nights)
may serve in part as a preventative mechanism of vapor leaving the ice sheet (*Berkelhammer et al.* 2016).



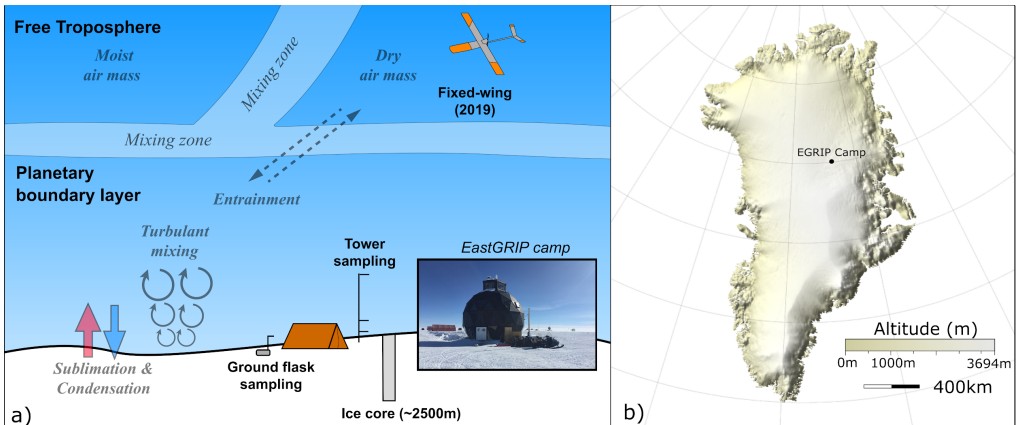

**Figure 1: a) Overview of the local hydrological cycle (excluding precipitation events) at the EastGRIP ice core camp. The water isotope sampling projects at EastGRIP in 2019 included ice core drilling, surface snow/ice sampling, continuous measurements of atmospheric air from a ~7m tower, and UAV sampling of atmospheric air) b) Location of the EastGRIP ice core camp on the Greenland Ice Sheet at 75.6°N 35.9°W.**

At a constantly varying height above the ice sheet (10s to 100s of meters in summer, lower in winter), a mixing zone between the surface and the PBL-free troposphere boundary allows for entrainment of water vapor from the free troposphere into the PBL. This exchange is not well understood due to the inability *thus far* to make measurements across the full PBL (*Boisvert et al.* 2016). The inclusion of outside air parcels is mediated by synoptic changes in atmospheric general circulation (*Schuenemann et al.* 2009). Characterization of these synoptic scale changes have been shown to be important to large scale melt events, such as the 2012 event across the Greenland Ice Sheet where changes in atmospheric circulation resulted surface melt (*Hanna et al.* 2014). Due to the conservation of water isotopes through mixing, gradients in water isotopes across the PBL-free troposphere-mixing zone may provide evidence of the amount of water vapor exchange between air parcels. As UAV methodologies improve, it will eventually be possible to provide constraints on net exchange of water vapor across the PBL-free troposphere interface.

**2.3 EastGRIP tower measurements**

During our UAV field campaign, simultaneous measurements of water isotopes were continuously taken at several heights above the snow surface. The tower set-up used for these measurements was similar to the system described in *Madsen et al.* (2019). Four air intake inlets were installed at 0.5, 1.0, 2.0 and 7.1 meter height above the snow surface from which air was pumped to a Picarro L2140-i analyzer in a temperature-controlled tent ~15m away using an auxiliary pump.

In addition to documenting a diurnally varying water vapor isotope signal, the tower measurements have successfully been used to observe a gradient in the isotopic concentration in the lowest part of the PBL (*Ritter et al.* 2016, *Madsen et al.* 2019). This gradient has been used to argue that the exchange between the atmosphere and snow surface is driving the diurnal water



isotope variations. Extending beyond tower heights will allow for the observation of entrainment processes and a better understanding of the formation of the ambient isotopic composition.

**2.4 Fixed Wing UAV Flight System**

While at the EGRIP camp in 2018 the team performed a proof of concept for airborne sampling and surface analysis using a small remote controlled sampling package and a multi-rotor UAV (DJI S-1000, DJI, Inc.) The system was able to obtain data

and samples for analysis up to 400 meters AGL, but navigation and control was very problematic, due to proximity to the magnetic pole and batteries at low temperature limited flight times to less than 15 minutes. Knowing that sampling was possible and effective, we moved our attention to fixed wing platforms that fly longer, higher and are more stable to operate.

The S2 fixed-wing aircraft was the chosen platform for the 2019 campaign. The S2 is a modular, autonomous, aircraft

designed by Black Swift Technologies, LLC (BST) for science missions, based on simple to operate electric propulsion aircraft with a modular payload. It includes a lightweight composite airframe design (Figure 2). The S2 is capable of conducting fully autonomous flights in unimproved areas such as an ice sheet. The aircraft can adjust to changing wind conditions in real-time, ensuring a high degree of stability for predefined mapping or atmospheric sampling applications (*Elston et al.* 2015b). The technical specifications for the S2 are listed in Appendix C.


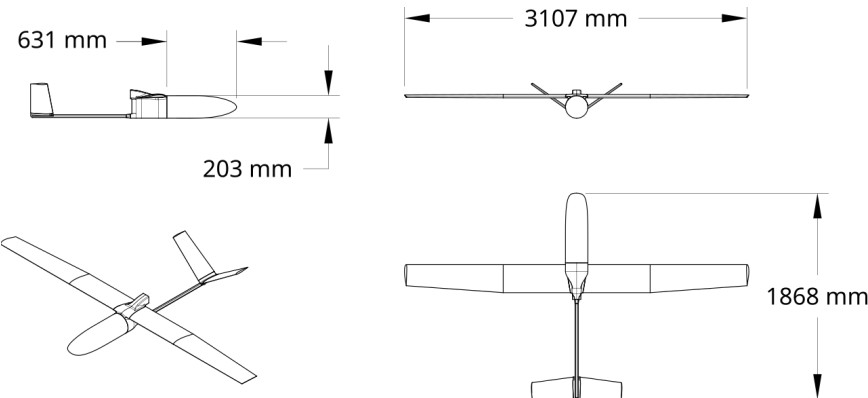

**Figure 2: Overview of the S2 fixed-wing aircraft.**

SwiftPilot™ (Black Swift Technologies, Boulder, CO) is a miniaturized autopilot system developed specifically for UAV applications, allowing for remote operation and autonomous operation monitoring with capability for intervention, and was

used in this study. Its modular CAN-bus architecture enables a large number of connectivity options, simplifying payload integration into the processing stream. Communication with the ground is enabled through the SwiftStation™ (BST), a portable tripod-mountable ground station (1.8 kg) that supports user-specific sensor payload integration, downlink, waypoint





programming and digital terrain model custom inputs, and operation control. The standard configuration, used in this study, contains a 3dBi gain 900 MHz dipole as well as a GPS antenna.

### 2.4.1 Nose Cone Sampling Pod

The flask sampling apparatus is contained within the nose cone, and a schematic of the system is shown in Figure 3. The payload is suspended on four carbon fiber rods spaced 140mm x 80mm apart which slide into the frame of the main aircraft where a manufacturer-supplied baseplate secures it in place with two spring-loaded latches. Eight glass flasks (Precision Glassblowing, Denver, Colorado) are suspended with memory foam in a series of modeled and 3D printed nylon-12 plates (KODAK Nylon 12). The printing was done on a XYZprinting da Vinci Super and sliced at a 15% hex infill with XYZware Pro. The glass flasks are approximately 550cc and include a supported dip tube to ensure the sample is adequately flushed during fill. A series of ¼" OD Bev-A-Line V tubing (Cole-Parmer) connects the glass flasks to a common inlet and outlet aluminum manifold (SMC, model SS073B01-08C) fitted with 12vdc solenoid valves (SMC, model S070B-6AC-M). Air samples are loaded into the glass flasks during a 5-minute flushing with air pulled from the intake port on the nose cone through the manifold and the selected flask to the diaphragm pump (KNF model DC-B 12V UNMP850). The pump is rated at 8 LPM but with altitude and system restrictions the flow rate is reduced to ~5 LPM, yielding approximately 50 flask volumes of flushing. Inlet and outlet valves are closed simultaneously so that flasks are not pressurized and remain at the ambient pressure of sampling. One of the extra valves is used for purging the manifold during sample analysis (see Section 2.6).

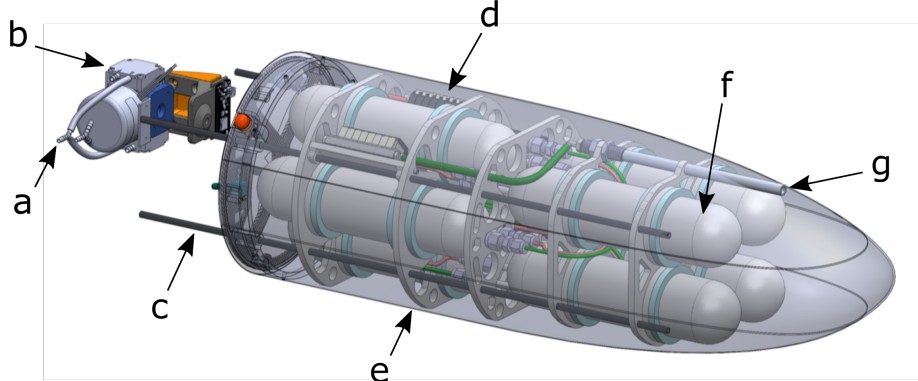

**Figure 3: Overview of the custom nose cone sampling pod. a) Air exhaust to outside the fuselage. b) 8 L/m KNF pump. c) Supporting rods for payload connection to UAV fuselage. d) 8-port valve manifolds to inlet and outlet. e) Nylon-12 baseplates with memory foam for flask suspension. f) 500 cc glass flasks with two port, dip tube. g) Air intake.**



### 2.4.2 Measurement Scheme

Similar sample pod control systems were used for both airborne and ground sampling. For sampling during flight, the on-board microcontroller (Adafruit Feather M0) works through the BST SwiftCore™ flight system to communicate to the ground station. Payload control is managed by a laptop with Linux (Ubuntu 18.04.2) connected over WiFi to the ground station. The microcontroller receives and manages commands to toggle valves and enable pumping. Environmental sensing is also fed into the BST SwiftCore™ and down to the ground station. The temperature and humidity is determined by an

E+E Elektronik EE03 sensor (±0.3°C and ±3%RH), and the pressure is determined by a high resolution (±1.5mbar) MEMS sensor (TE Connectivity MS5611).

In addition to measurements of samples taken during flights, a small (2m) sampling tower was used for flask sampling to provide an additional near surface data point and also allow an intercomparison with tower measurements of water vapor

isotopes at EastGRIP. On the ground, a second microcontroller was connected to the sample pod with a USB cable. Its tasks included controlled functions 1) flushing dry air through flasks prior to flight, 2) sample acquisition from the 2 meter tower, and 3) computer controlled release of samples for isotopic analysis. Flasks from both flights and ground sampling are introduced to an L2130-i Picarro instrument for isotopic analysis by opening a single port on the flask. Before air sample ports are opened, dry air is plumbed into a spare valve at the back of the manifold to push out atmospheric air left in the

manifold. Air samples are pulled into the instrument at a rate of 30 sccm through a tube from the common port of the valve manifold for approximately 12 minutes. As water vapor is introduced to the CRDS cavity, isotopic mixing with the previous dry air parcels can affect the instrument's response to new samples. To address this, the first 3 minutes of observation for any one sample is cropped from averaging. Additionally, the last 3 minutes are also cropped, defined from the time after a flask valve closes and the flow rate decreases. Cropping in this way also allows a mixing ratio/specific humidity to be determined

for calibration. Values for any one sample are determined from the average over approximately 6 minutes. For a systematic diagram of the drone and ground sampling, see Appendix A.

The methods insured equal treatment of samples collected in-flight or on the ground. This served two purposes, 1) to establish the isotopic bottom end-member of the vertical profile. and 2) to enable the comparison of the sample pod

measurements with the established *in-situ* tower measurements of water vapor concentration and isotopes (Picarro L-2140-i), taken at the same time within a distance of 10 m.

### 2.5 Water Vapor Isotope Measurements and Calibration

Systems have been developed by numerous groups to calibrate Picarro CRDS instruments used in continuous flow applications (*Gkinis et al.* 2011, *Steen-Larsen et al.* 2014, *Jones et al.* 2017a), and each represents an evolution in design and

performance. Due to the proven success with multiple measurement campaigns completed on ice cores with the calibration

setup described in *Jones et al.* (2017a), we used the same principles in this setup for the calibration of the system in the field. It meets the ideal criteria for a calibration system as described in *Bailey et al.* (2015), that includes a) enabling the introduction of low volume mixing ratios for calibration, b) mitigating standard drift, and c) utilizing multiple water standards in the calibration scheme. The system schematic is shown in Figure 4.


Inlet system to introduce water standard and samples to CRDS System

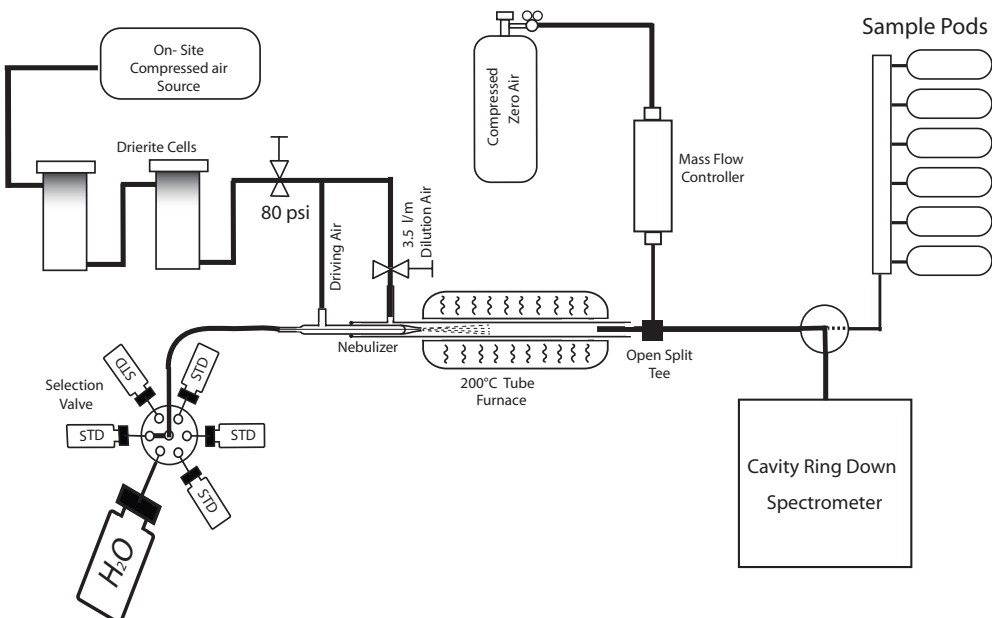

**Figure 4: System diagram of the inlet system that introduces water vapor from a suite of isotope standards, or from glass flasks in the nose cone sample pods.**

A Valco six-port stream selector valve (Valco Instruments Co. Inc.) controls selection of water standards in 30 ml Pyrex

glass vials fitted with a 1/16" capillary and a pig-tail vent tube. The selected water standard is introduced to the flash evaporator system through a concentric nebulizer (Meinhard, TL-HEN-150-A0.2), powered by high-pressure (80 psi) dry air. The nebulizer inducts the water at 160 - 250 uL/min and converts the liquid to a fine spray of approximately 1.5μm droplets inside a 20 cm x 1.8 cm diameter Pyrex tube heated to 200° C by a surrounding ceramic tube furnace (Whatlow, VC400N06A). The spray is mixed with a separate flow of compressed dry air at 3.5 L/min to achieve a vapor concentration

of approximately 20,000 ppm. At the end of the furnace tube is an open-split style intake line (Swagelok 3.175 mm OD x 2 mm ID x 10 cm stainless-steel tubing) inserted approximately 5 cm into the Pyrex furnace tube. Excess water vapor from





the open split then vents to the room. An additional dry air (<50 ppm $H_2O$) is then introduced through a mass flow controller (Alicat Scientific, MC-100SCCM-D/5M) into the output line to further dilute water vapor down to desired concentrations necessary for calibration. At this stage, a manual 3-way valve selects either the vapor output of the calibration system, or

selected glass flasks of the sample pod, to enter the CRDS system. Control of the sample pod valves is coordinated with the microcontroller and the CRDS computer. The flow rate into the CRDS analyzer is approximately 30 sccm and controlled by a critical orifice inside the instrument and a pump (Vacubrand MD1) attached to the Picarro L-2130*i*.

Raw values from the CRDS system are corrected in post processing and tied to known values of isotopic water standards and

corrected for the instrument's response to humidity. This is required because our atmospheric water vapor samples (typically <5,000 ppm $H_2O$) are outside of the standard operating range of the Picarro L-2130i, which is optimized for the analysis of liquid water samples (10,000 to 25,000 ppm $H_2O$). Counting statistics for CRDS instruments are heavily dependent on sufficient concentration of gas species warranting calibration across a range of humidities and isotope standards. The isotopic water standards and their uncertainties are given in Table 1.


**Table 1: Tracing of uncertainties is provided for primary reference water standards (\*) and secondary water standards developed in the laboratory and are reported in units of per mil. The four secondary standards (BSW, ASW, PSW, and SPGSW) are previously calibrated in the laboratory and are defined relative to the primary standards (VSMOW2, SLAP2, and GISP) on which values and uncertainty are reported by the IAEA. Secondary standards are reported with uncertainty determined across multiple**
**IRMS and CRDS platforms. In parenthesis is the combined uncertainty of both the primary and secondary standard tie, added in quadrature.**

| Standard | δD (‰) | δD uncertainty | $\delta^{18}O$ (‰) | $\delta^{18}O$ uncertainty |
|---|---|---|---|---|
| VSMOW2* | 0 | 0.3 | 0 | 0.02 |
| SLAP2* | -427.5 | 0.3 | -55.5 | 0.02 |
| GISP* | -189.5 | 1.2 | -24.76 | 0.09 |
| BSW | -111.65 | 0.2 (1.3) | -14.15 | 0.02 (0.10) |
| ASW | -239.13 | 0.3 (1.3) | -30.30 | 0.04 (0.10) |
| PSW | -355.18 | 0.2 (1.3) | -45.41 | 0.05 (0.11) |





| SPGSW | -434.47 | 0.2 (1.3) | -55.18 | 0.05 (0.11) |
|---|---|---|---|---|

To characterize the instrument's isotopic response to different water vapor concentrations, suites of measurements for each water standard are made under a range of humidities, from 500ppm to 25,000ppm water vapor. This is accomplished on the

system by adding measured amounts of additional dry air to the open split vaporizer that feeds the instrument. Dry air was provided by one of two sources: a dry air generator (Model CDA-10 by Altec Air, Broomfield, Colorado) that produced 10 lpm air at -73°C dew point; or dry air from a size 300 compressed air tank (Zero Grade, AIRGAS, USA). Both were capable of supplying air with less than 50 ppm $H_2O$. A mass flow controller (Alicat model MC-100SCCM-D) metered dry air to achieve a suite of desired humidities for calibration purposes. The resulting data were used to create an interpolated surface

(Hermit Interpolation, Mathematica) of measured vs. adjusted, or true isotopic values.

This calibration procedure was done several times throughout the 2019 field season to capture long-term instrument noise in response to humidity. Atmospheric samples were calibrated to the set of humidity measurements closest in time, ranging from as long as 7 days apart but typically 1-3 days throughout the season. *Steen-Larsen et al.* (2013) indicates that

correctable linear drift may occur local in time to the measurement period due to strong diurnal temperature changes around the instrument. Because humidity calibrations were not regular about each measurement at the time scale of diurnal temperature change, the correction was not performed in this study. Future campaigns will include a higher calibration density to account for this.

**2.6 Uncertainty in Sampling and Intercomparison with On-Site Water Vapor Tower**

Outside of CRDS instrument performance, the UAV sampling system itself introduces sources of error. This uncertainty is associated with acquisition and transport of the sample water vapor as well as environmental change during the flight period. To understand the uncertainty in captured water vapor during the 2019 season, two different flask pod intercomparisons were performed in conjunction with the separate 2-meter tower-isotope setup detailed in Section 2.3. For the intercomparison, each of the six flasks from three different sample pods was flushed with air from 2-meter altitude for 5 minutes. As a total of

eighteen flask measurements correspond with an hour and half of sampling, this test is sensitive to changes in atmospheric water vapor isotopic composition. A more appropriate test would be to produce standardized water vapor as described in Section 2.5 and sample from that stream. This is challenging because the most accurate test would be to produce water vapor at a rate that can match the 5 LPM sampling throughput of the pump, which is currently unachievable due to limited amounts of water standards. Though sampling was performed over this longer period of time without standard water vapor, the

highest 1σ standard deviations of any one pod was of 0.45 in $\delta^{18}O$ and 2.80 in $\delta D$. These values can be seen as the pessimistic view of uncertainty due to the non-ideal sampling situation, but are reasonable given that previous uncertainty





estimates on in-situ water vapor isotope measurements range from 0.14 per mil in $\delta^{18}$O and 0.85 per mil in $\delta$D (*Steen-Larsen et al.* 2014) to 0.23 per mil in $\delta^{18}$O and 1.4 per mil in $\delta$D (*Steen-Larsen et al.* 2013) depending on the environmental conditions.


A comparison of UAV and tower deuterium excess (dxs) data is shown in Figure 5. Deuterium excess is defined by *Dansgaard* (1964) as dxs = $\delta$D − 8$\delta^{18}$O. The dxs is a more sensitive intercomparison metric than $\delta^{18}$O and $\delta$D and will more clearly show discrepancies between different measurement schemes. An intercomparison was done at four different times: 1) during 2 m sampling during two different flights and 2) during two different pod intercomparison measurements at 2 m.

There is general agreement for dxs across the two platforms with a slightly more positive value for the UAV-isotope system.

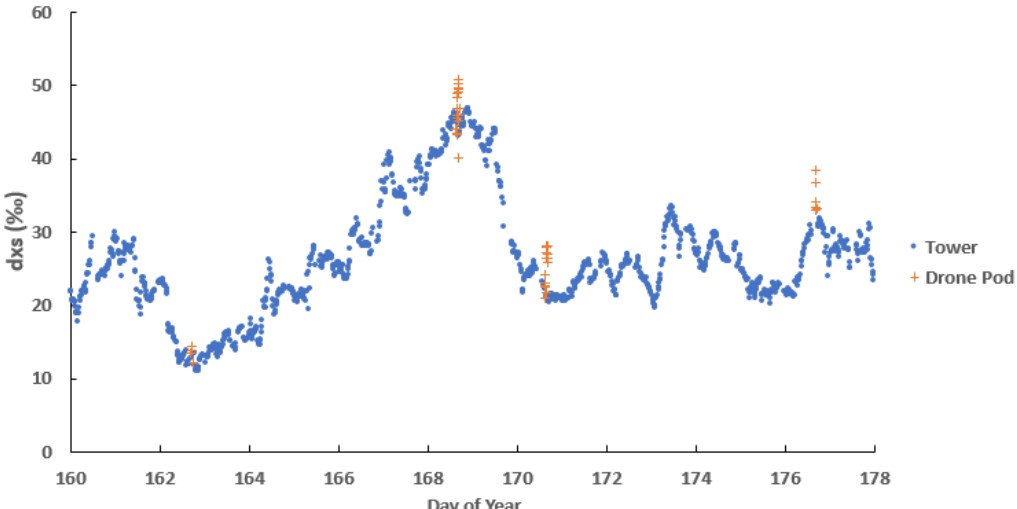

**Figure 5: Comparison in second order parameter dxs between the tower setup present at the EastGRIP camp in 2019 and 2 meter pod intercomparison measurements. Drone pod measurements on 163 DOY (Day of Year) and 176 DOY correspond to flask measurements taken at 2 meters during a flight mission. All measurements from both tower and UAV are tied to the same isotopic**
**water standards listed in Table 1.**

**2.7 Boundary Layer Prediction**

During the 2019 field campaign, we used environmental measurements (pressure, potential temperature, specific humidity) taken in real-time during each flight to evaluate Euclidean distance in the measurement domain to infer where the PBL/free troposphere transition occurs in the spatial domain. The results were used by the pilot to make in-flight decisions about

sampling altitudes for isotopic analysis. After the 2019 field campaign, we explored additional PBL identification algorithms. The PBL and free troposphere are largely decoupled, allowing for cluster density evaluation to determine the PBL height (*Krawiec-Thayer* 2018). As the PBL structure varies in shape and magnitude for any one observational parameter, other methods such as gradient interpretation of single environmental variables are less useful (*Krawiec-Thayer*

2018). The most promising algorithm, the Calinski-Harabasz criterion index (CHCI), is explained in Appendix D. The global

maximum of this index is assumed to be the height of the PBL. The Calinski-Harabasz criterion index will be utilized in

future field campaigns to detect the PBL in real-time during flight in addition to user judgment. An example of the CHCI for

PBL height determination is shown in Figure 6.

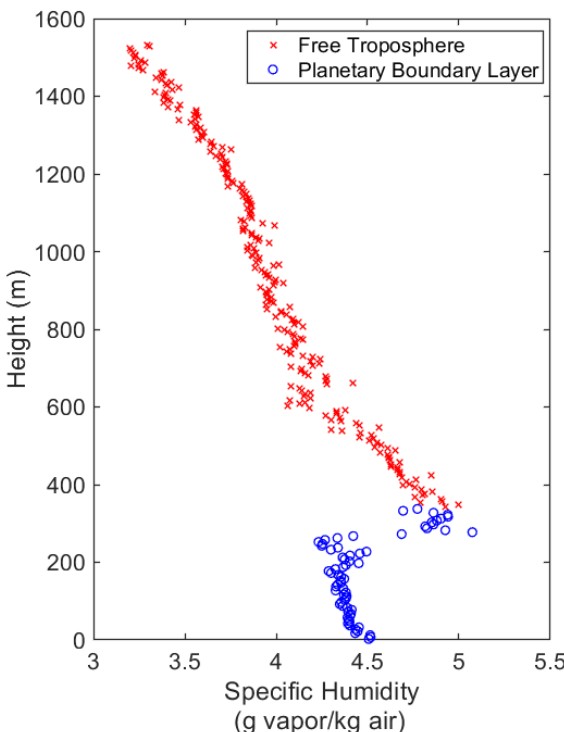

**Figure 6: The Calinski-Harabasz criterion index (CHCI) applied to the sampling flight on June 12th with *a-priori* assumption of**
**K=2. Groups are assumed to represent the free troposphere and PBL, though more structure may exist. The boundary at ~325 m**
**using CHCI is not in agreement with that determined using the Euclidian distance in Figure 7, which shows a likely boundary at**
**~275 m. Figure 10 shows an example of a failure in Euclidian distance to predict the boundary layer. The use of CHCI improves**
**the PBL prediction algorithm, as determined in this study.**

**2.8 Typical analysis day and sample acquisition**

Before flight is considered, the local weather is evaluated to determine the potential for mission success. To prevent potential

icing, a nearby ceilometer (Vaisala Ceilometer CL31, Vaisala, Boulder, Colorado) present at the EastGRIP camp was used to

safely determine that cloud cover was significantly higher than the highest flight altitude in the flight plan. Flights were not

performed during precipitation events. Acceptable wind speeds were considered less than 10 m/s, two-thirds of maximum

wind operation of 15 m/s for the Black Swift S2 aircraft.






For any given analysis flight, a sequence of steps are completed to ensure quality control: 1) Calibration of the water isotope measurement system (Section 2.6), 2) On going isotopic measurements at a 2-m tower during the flight (Section 2.7), 3) Identification of the PBL during flight using real-time temperature and R/H from the aircraft (Section 2.8), 4) Atmospheric sample acquisition during flight, and 5) Isotope measurement following the flight, in a heated field tent (Section 2.6).


A calibration of the Picarro L-2130i is performed close to the time of flight. Before a flight, both ground-based and UAV-based glass flasks are flushed with dry air (75 ppm water vapor) for 10 min. Before launch (time permitting), an extra 2 m measurement is taken with the ground sampling system detailed in Section 2.5.2. After launch, the pilot ascends at an autopilot-controlled rate of 2 m/s in a circular pattern (a 68 m diameter orbital). The ascension rate can be affected by local

wind speeds requiring a slower vertical climb than the UAV is otherwise capable of. While a faster ascension is possible, a slower climb also minimizes hysteresis for the atmospheric sensors onboard the UAV. At the top of the climb, the aircraft automatically enters a holding orbital pattern at constant altitude while the operator assesses the real-time algorithmic determination of the PBL. The operator then inputs the altitude of the sampling locations for water isotopes above and below the PBL.


The UAV then descends to the first/highest sampling altitude. At each sampling altitude, the pilot initiates flask sampling. The sample procedure can be broken into three steps: 1) idling, 2) flushing, and 3) equilibration. When the UAV reaches the first sampling altitude, the UAV will maintain altitude (idle) for approximately one minute to eliminate hysteresis of the environmental sensors. The diaphragm pump is then turned on and each port on the flask is opened for a three-minute flush

of ambient air to address memory effects on the interior glass surfaces. Then, the pump is turned off in order for the flask to equilibrate to ambient pressure for 10 seconds. Finally, the valves are closed, and the process is repeated for a second flask, providing paired measurements at each altitude. The nose cone sampling pod holds 8 flasks, allowing for paired measurements at four altitudes. However, due to battery limits on site, the payload was generally flown with 6 flasks (3 pairs). The aircraft is then directed to land. Both the UAV atmospheric samples and ground-based samples (from 2 m height)

are then analyzed on the water isotope measurement system and calibrated to the most recent system calibration (Section 2.5).

**3 Results and Discussion**

**3.1 Retrieval of Water Vapor Isotopic Composition about the PBL**

Though CRDS measurement of water vapor isotopes by aircraft is not new (Section 1), its capture and retrieval by UAV for

later measurement is novel. Arctic environments present major logistical challenges for fieldwork. The remoteness of field camps, such as EastGRIP, makes logistics challenging and limits the amount of field personnel. The potential for extreme



weather, cold temperatures, blowing snow, and safety are all significant factors that limit scientific outcomes. For these reasons, even the most careful planning will still result in some unforeseen challenges. During our field campaign, we realized that we had to improve system sampling turnover time to produce more flights per day, that hysteresis in the

environmental sensor could produce artifacts in PBL detection, and that our 2-3 person field crew was inadequate to have good diurnal sampling coverage since all people slept during the same hours. A larger team would have provided an option for day and night shifts as there were 24 hours of sunlight during the field campaign.

Despite unforeseen challenges, we achieved a total of six sample-taking flights from June 3rd to June 26th, 2019. An

example of environmental sensor data for June 12th is shown in Figure 7. We found varying amounts of structure in isotope space across all six flights (Figure 8). Large transitions between water vapor isotope surface measurements at 2m and values above and below the PBL/free troposphere (FT) transition are apparent in the June 6th and June 12th flights. The June 12th flight, in particular, exhibited the largest changes in water isotopes with altitude AGL (Figure 8). The other four flights in contrast had little variability, suggesting that the PBL was unstable (i.e. well mixed). *Berkelhammer et al.* (2016) suggested

that summertime nights at Summit, Greenland would present the conditions for stable stratification of the atmosphere, but that this claim was unprovable using towers alone. The June 6th flight, which happened at 1:30am (a summertime night), provides some evidence in support of the *Berkelhammer et al*. (2016) claim, but we do not have enough data yet to elaborate further. In 2022, we will use an improved UAV-system setup to generate a comprehensive diurnal data set spanning many weeks worth of time.

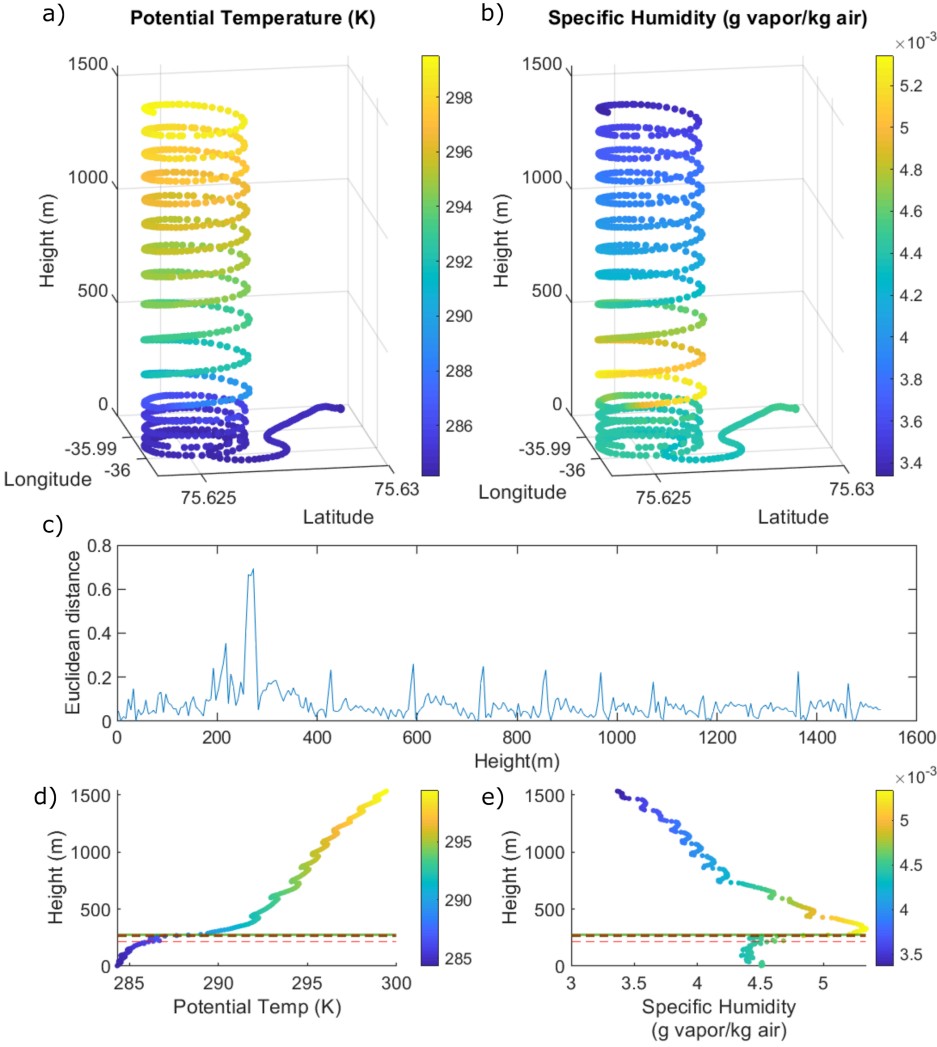


**Figure 7: The ascension profile for the June 12th flight mission available to the operator to determine PBL location. The flight path reached a maximum altitude of ~1500m AGL (a and b). The potential candidate for the Euclidean distance determination of the PBL is shown to be at approximately 272.5 meters ABL (c). The top three candidates for PBL all correspond to approximately the same location (green and dotted lines d-e). There is a modest gradient in potential temperature over the flight path of about 16°K (d). Specific humidity shows an inversion in the first few hundred meters of flight at the determined location of the PBL (e).**




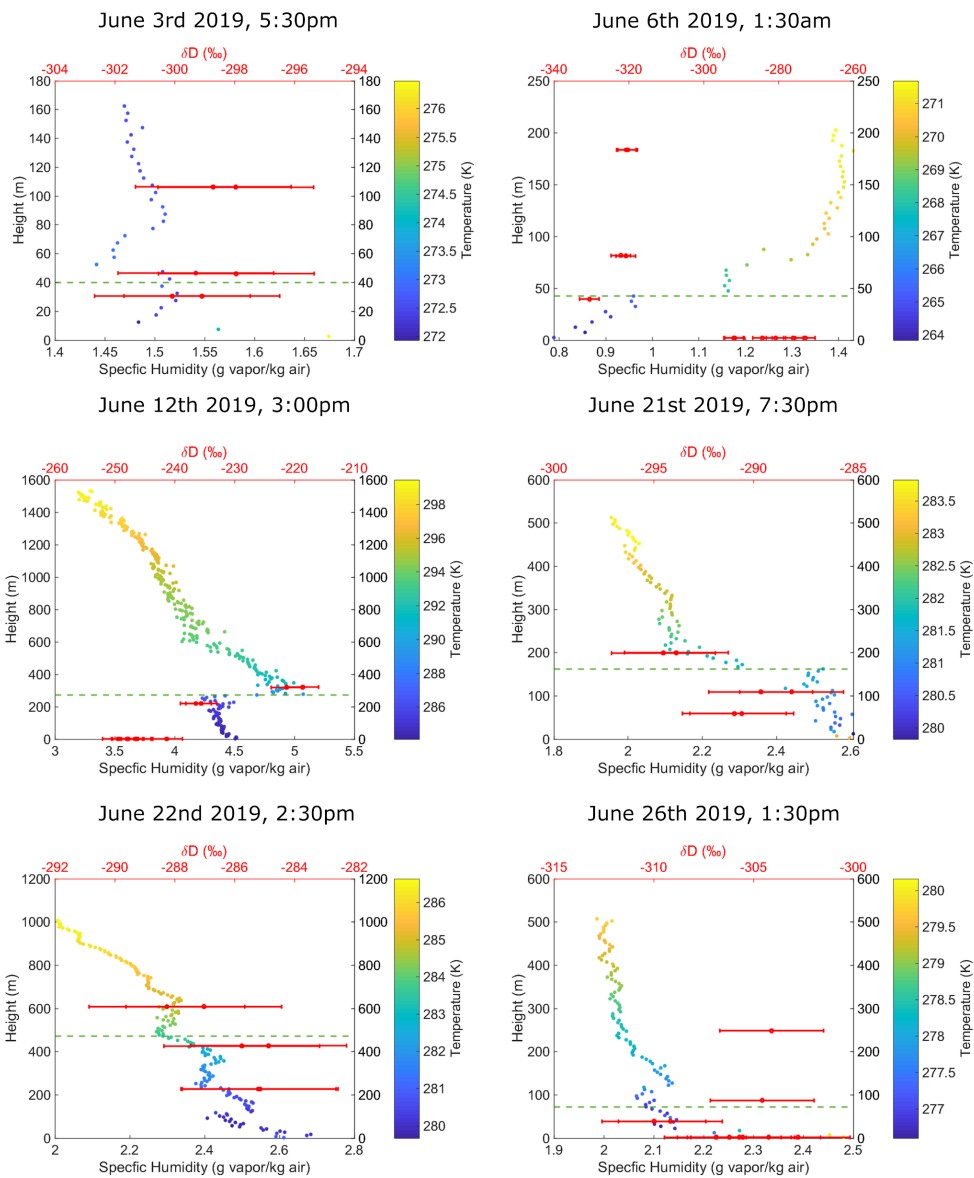

**Figure 8: The six flights during the 2019 summer field session at EastGRIP field camp. Each flight includes a specific humidity and temperature measurement, which is binned for the values at each altitude during the ascension and descent. The result of the determination of the PBL by the operator outlined in Section 2.7 is plotted as the dashed green line. The resulting isotope measurements at sampled altitudes are shown in red. Error bars are determined from flask intercomparisons (Section 2.6). Additional plots of $\delta^{18}O$ can be found in Appendix A.**






### 3.2 Hysteresis and Calinski-Harabasz Criterion Index (CHCI) and PBL Detection

The CHCI was calculated post-flight for comparison with 1) the self-similarity of Euclidean distance (used during the 2019 field campaign, but later updated to the CHCI approach) and 2) operator determination of the PBL. The results are shown in

Appendix A. The CHCI had a direct match with Euclidean distance for half of the flights. In the other half, the CHCI predicted altitudes significantly higher than the other determinations. The results of our comparison reveal that our original PBL-detection algorithm using Euclidian distance needs improvement (Figure 9). Specifically, we have determined that Euclidean distance can under or overestimate the height of the PBL due to sensor (temperature and humidity) hysteresis. This hysteresis exceeded the stated manufacturer response time for the atmospheric temperatures we encountered, discussed

in Appendix B. The hysteresis is almost entirely a result of the rate of ascent during flight. Before a flight, the UAV is static at ground level, thus temperature and humidity measurements will be stable, varying only slightly with small changes in surface conditions. The energetic pneumatically-driven launch of the aircraft (a 12 G force) results in a rapid increase in altitude that can introduce a bias into the sensor output due largely to the thermal mass of the sensor and slow response to rapidly changing conditions. A similar effect occurs anytime the rate of ascent is not constant, such as when the UAV

transitions between different orbitals (i.e. a sampling orbital and landing orbital).

A case study in Figure 9 illustrates a shift in orbitals from the June 21st mission. The operator moved from the initial launch orbital to a lower altitude to begin an ascension profile. During the transition between the two orbitals, the aircraft moved from about 110m to 60m in altitude in ~1 minute. During the transition and immediately during the ascent, multiple

temperature and humidity values were generated for the same altitude creating a region of varying hysteresis effects that can bias PBL prediction by Euclidean distance, ultimately causing the operator to misidentify the altitude of the PBL. More concisely, the algorithm detected this data anomaly as atmospheric structure, when in fact it was due to hysteresis. While removing this skewed data could be an easy fix, the stabilization of temperature and humidity to that new starting altitude biases the beginning of the climb just as it does at the surface before launch.

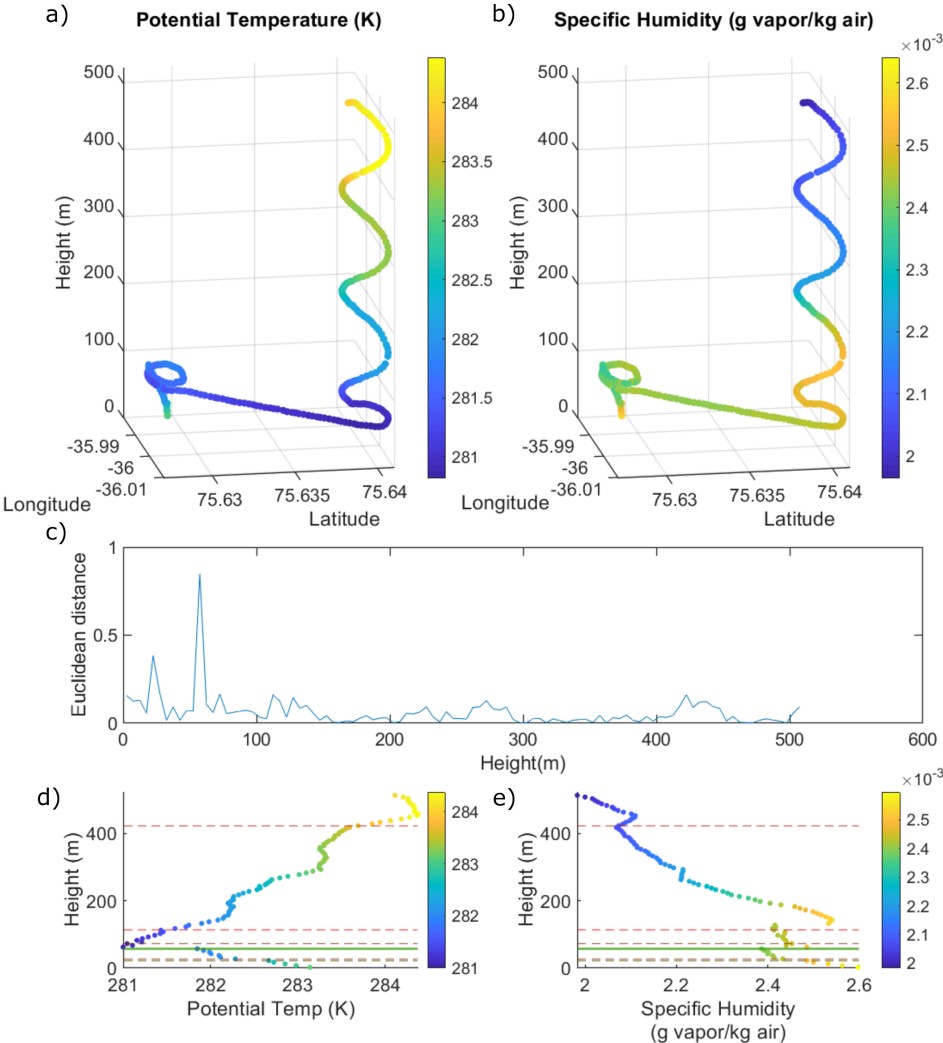


**Figure 9: The ascension profile for the June 21st flight mission available to the operator to determine PBL location. The flight reached a maximum altitude of ~500m (a and b). Post launch, the operator flew from the initial launch at ~110m AGL to ~60m AGL over the course of ~1 minute. Settling of both temperature and humidity due to hysteresis during that time was flagged incorrectly by Euclidean distance (c). The resulting predictions of PBL locations (solid green line for the most likely, dashed line**
**for the next four likely, d-e) are scattered across the space.**

The hysteresis effect is also noticeable in the CHCI (Figure 10, green circles). Relaxing the *a priori* assumption of a single

PBL that separates the surface atmosphere from the free troposphere, additional transition regions can be identified. As





CHCI uses Euclidean distance to establish variances, it is also subject to potentially poor predictions in situations of significant hysteresis. However, its ability to establish regions of similarity, such as the case of the transition region between

launch orbital and the ascension orbital during the June 21th mission provides an objective method of informing the operator of potential false positives for the boundary layer altitude. In this specific case, three of the top five PBL altitudes predicted by the Euclidean distance algorithm can be flagged as incorrect. However, even with sensor hysteresis, we determine the CHCI to be an effective tool to assist in fast mid-flight evaluation of the boundary layer by the drone operator.

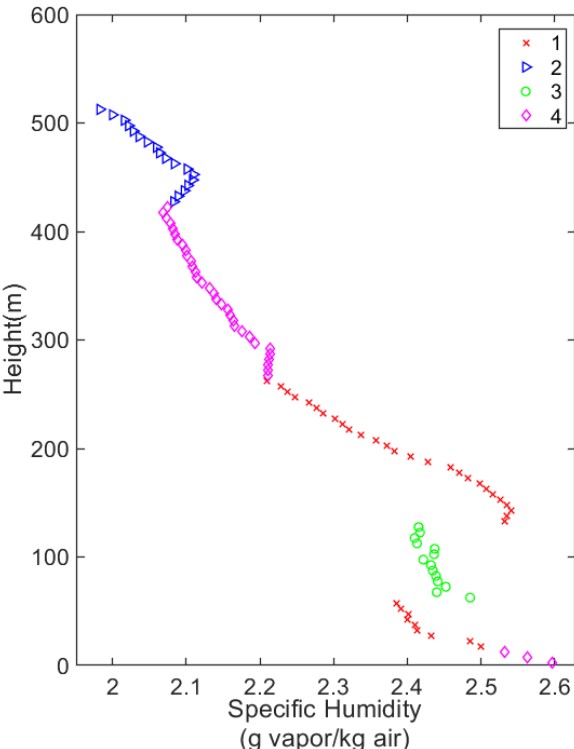

**Figure 10: Specific humidity over the ascension for the June 21st flight partitioned into groups by the CHCI with the *a-priori* assumption of K relaxed from 2 to 4. The region of transition the operator took post launch between ~110m and ~60m is clearly evident as a separate group (green circles). In cases where artificial structure exists due to sampling patterns, CHCI may assist the operator by flagging those areas.**

Overall, there are two options for overcoming the effects of hysteresis: 1) better sensors and 2) changes to flight mission

plans. We have identified the Vaisala RSS-421 sonde sensor to meet the first requirement. The RSS-421 includes a low thermal mass fine-wire thermocouple and heated humidity sensor with bakeout unit, which will allow for faster response in arctic conditions. This sensor has already shown to be capable of producing accurate temperatures in challenging UAV



fixed-wing missions (*Frew et al.* 2020). For flight planning, relocating launch sites to be as close to the ascension orbital as possible will reduce hysteresis during horizontal transitions between orbitals. The ascension rate can also be slowed to less

than 2 m/s allowing the maximum time for sensors to equilibrate with the surrounding atmospheric conditions. The tradeoff is that this may require reducing the maximum flight altitude to conserve battery life and reduce the bank angle. A sharp bank angle decreases the lift coefficient (*Williamson* 1979), and a higher angle of attack is needed to maintain ascension rate in tailwind situations (*Blakelock* 1991). When the pitch angle needed is too high and outside the flight envelope, the Black Swift Technologies autopilot will slow ascension to protect the aircraft. It is assumed that variability in temperature,

pressure, and humidity is small in the x and y plane, allowing for a large increase in orbital diameter to reduce bank angle significantly.

## 5 Conclusions

We have presented a UAV-isotope sampling platform and methodology capable of measuring atmospheric water vapor and its stable isotopes within the planetary boundary layer (PBL) and lower troposphere in a polar environment. We utilize a

fixed-wing UAV (Black Swift Technologies) with flight times in excess of 45 minutes with the capability to reach 1,600m AGL. Multiple nose cones allow for collection of air in 8 glass flasks, enclosed within a 3D printed support structure that critically withstands 12Gs of force during takeoff. In this study, the total system is used to sample above and below an algorithmically-detected PBL, with the ultimate goal of improving our understanding of water molecule exchange between the ice sheet, PBL, and lower troposphere. The UAV-isotope sampling platform was deployed at the EastGRIP ice core field

site in summer 2019 and its success signifies a novel new platform for research.

Atmospheric air is collected during flight within glass flasks contained within the nose cone of the UAV, which is subsequently measured on the ground for water vapor and water isotopic content ($\delta D$ and $\delta^{18}O$). Our sampling technique is precise down to 2.8‰ in $\delta D$ and 0.45‰ in $\delta^{18}O$. This is a worst case scenario and it is likely the measurements are more

precise, as we have based the uncertainty on the reproducibility of 18 measurements of atmospheric air from 2m height on a tower; this air is subject to diurnal variation and local weather variability over the course of 90 minutes. However, the isotopic uncertainty presented here is on the order of previous surface science publications (*Steen-Larsen et al.* 2014) and within an order of magnitude of modeled surface dynamics (*Madsen et al.* 2019). Furthermore, airborne UAV isotope measurements are critical in evaluating regions of the atmosphere not easily accessible by remote sensing or large aircraft.

Due to the ability to sample an individual flask for an extended period, drone sampling significantly outperforms large aircraft isotope profiles in precision (*Herman et al.* 2014, *Schneider et al.* 2015). For remote sensing, *in-situ* UAV sampling allows for robust benchmarking that would otherwise be difficult or impossible to obtain.

During a 2019 field campaign, we sampled atmospheric air above and below the PBL. An algorithm based on Euclidean

distance was initially used to determine the height of the PBL. We find that prediction of the PBL altitude is more difficult





when the rate of ascent during flight exceeds the ability of the sensor to accurately measure environmental variables like temperature and humidity. The end result is sensor hysteresis that introduces artifacts in the PBL detection. Eliminating or identifying these artifacts is critical for future field campaigns. We have subsequently improved the algorithm to also include the machine-learning index, the Calinski-Harabasz criterion index (CHCI), for rapid mid-flight decision-making.

This algorithm allows the Pilot-In-Command to make determinations in real-time about the height of the PBL, as well as which altitudes to sample atmospheric air based on this information. For future flight missions, we will carefully regulate the rate of ascent and include better performing temperature and humidity sensors with minimal time constants, all of which will reduce hysteresis for PBL detection.

Our field campaign in 2019 resulted in the first measurements of atmospheric water isotopes above and below the PBL on the high-altitude Greenland Ice Sheet. Across 6 sample-taking missions, we observed significant variation in structure in water isotopes on either side of the PBL. This forms the basis for future campaigns to collect high-temporal density measurements (flights every 4-6 hours across many weeks) at key missing scales that will improve ice-to-atmosphere modeling and mixing processes, ice sheet mass balance, satellite detection algorithms, moisture tracking, ice core science,

and modeling the hydrologic cycle in general.

A field campaign for return to EastGRIP is scheduled for summer 2022. Future improvements to the UAV-isotope system will be primarily focused on logistical improvements that increase the number and frequency of flights. Additional flight crew will be available for nighttime flight missions. To ensure a balanced diurnal flight schedule over weeks of time, with

the goal of one flight every 4-6 hours, a precessing schedule of calibration times will be used. Each calibration will be done every 2-4 days, lasting 12 hours, starting at different times of day. This ensures that we do not consistently lose the ability for UAV sampling at the same time for every calibration, e.g. from 12pm-12am. The combination of these improvements will allow the potential maximum number of flights per day to increase from two to as many as six, while balancing the timing of calibration. Additional improvements will include higher-quality, fast-response temperature and humidity sensors

on the aircraft, along with a lighter pump and manifold system that should allow greater flight time. Beyond Greenland, this platform is readily adaptable to other scientific disciplines, and will be used in an upcoming permafrost project to measure atmospheric methane emissions and soil moisture content in Alaska.



**Appendix A: Additional Schematics and Figures**

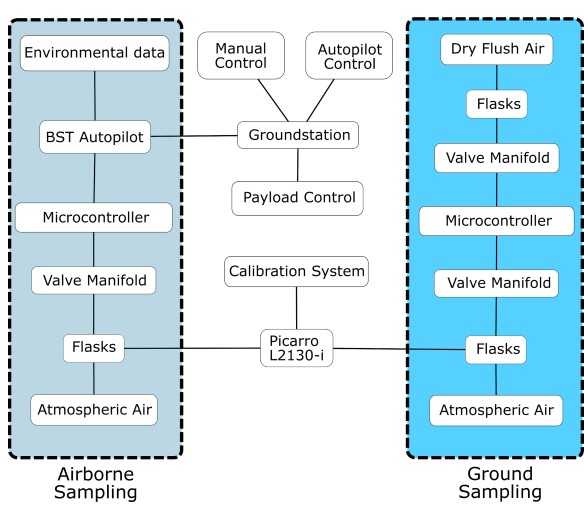

**Figure A1: UAV-isotope system diagram showing control and sample exchange between airborne/ground sampling and measurement subsystems. Both ground and airborne sampling are performed identically though their control methods differ.**

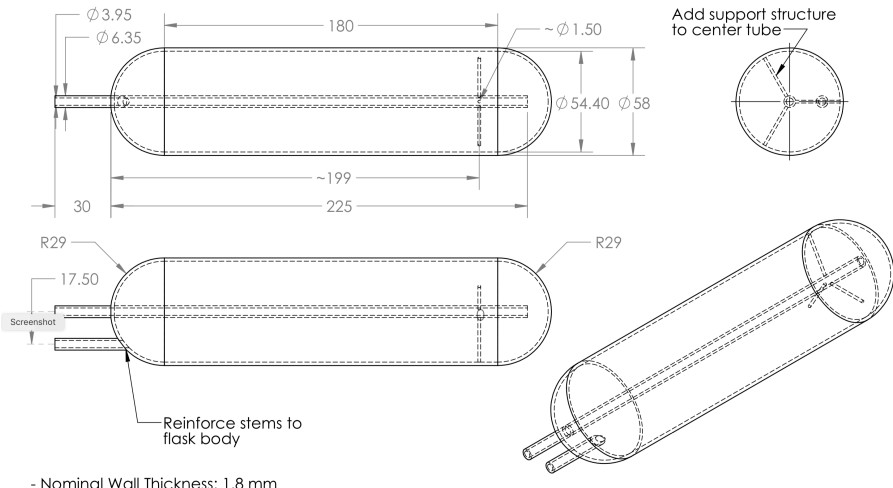

**Figure A2: Design document for the glass flasks onboard the S2 payload.**



**Figure A3: The six flights during the 2019 summer field session at EastGRIP field camp. Includes collected environmental data during flight for both ascent and descent as well as measured isotope values for oxygen.**

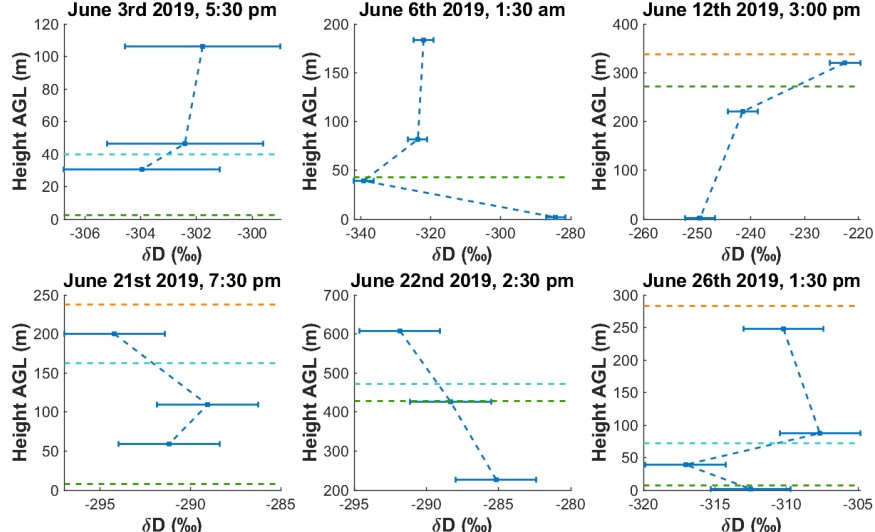

**Figure A4: A comparison between different methods of determining the location of the PBL. The location predicted from the first off-diagonal of Euclidean distance is indicated by the green dashed line. When the CHCI prediction is different, it is plotted in orange. When the operator determined a different location of the PBL, it is plotted in cyan. To illustrate the structure observed relative to these predictions, average isotopic values of δD for flasks taken at each height are shown in blue.**





**Appendix B: Hysteresis Correction**

A supplier listed hysteresis curve was used to correct for the capacitive humidity sensor (HC103M2) onboard the ee03 sensor used on the Black Swift Technologies S2. Note that observed hysteresis was much greater than this.

| Temperature (°C) | Response Time (sec) |
|---|---|
| 20 | 0.56 |
| 0 | 0.94 |
| -20 | 5 |
| -40 | 29.4 |
| -60 | 190 |

The correction was made by linearly interpolating a function (MATLAB fit() function) with the above values to determine

measured time vs. true time then applied to the altitude that represents the new time for the measurement.

https://sensortech.hu/pdf/EE/HC103M2-adatlap.pdf

A supplier listed response time for temperature measurement of the ee03 sensor was not available within the temperature ranges measured within the study and assumed to be negligible.




**Appendix C: The S2 Drone**

Scientific missions the S2 has flown prior to this study include mapping soil moisture with a radiometer (*Dai et al.* 2016), a calibration mission including a 12-band multispectral camera system (*Wang et al.* 2016), measuring snow-water equivalent with a radiometer (*Yueh et al.* 2018), and a volcano sampling mission that involves difficult operations into the plume of an

active volcano (*Wardell et al.* 2017). The S2 is currently in use by the National Oceanic and Atmospheric Administration (NOAA) for wildfire applications (*Gao et al.* 2017) and it has flown in various challenging environments including at high altitude during atmospheric sampling campaigns in the San Luis Valley in Colorado (*de Boer et al.* 2018). The S2 is designed for operations at altitudes up to 6,000 m AMSL in support of The National Aeronautics and Space Administration (NASA) science missions (*Elston and Stachura* 2017).


The S2 utilizes the SwiftCore™ Flight Management System for avionics control, communication, and command, designed by BST. It comprises the SwiftPilot™, SwiftStation™, and SwiftTab™ user interface, along with support electronics. SwiftTab™ runs on Android devices like smartphones or tablets. Flight plans 1) can be uploaded, created, and modified before and during flight, 2) can use georeferenced data points for systematic surveying including pre-defined banking and

spirals, and 3) are fully autonomous from launch to landing. Immediate preliminary analysis and decision making is supported via real-time telemetry and control capabilities.





**Table 1: Black Swift Technologies S2™ specifications**

| | |
|---|---|
| **Mission** | |
| Ingress Protection (IP) | IP42 |
| Launch Mechanism | Pneumatic launcher |
| Flight ceiling | 6,000 m AMSL |
| Maximum stable wind speed | 15 m s$^{-1}$ |
| **Flight** | |
| Stall Speed | 12.0 m s$^{-1}$ |
| Cruise Speed | 19.0 m s$^{-1}$ |
| Takeoff Speed | 20.0 m s$^{-1}$ (no flaps) |
| Landing Speed | 16.5 m s$^{-1}$ (full flaps) |
| | 19.0 m s$^{-1}$ (no flaps) |
| Roll | ± 45° |
| Pitch | ± 20° |
| Take-Off / Landing Corridor | 200 m x 15 m |
| Endurance | 120 min maximum |
| | 090 min nominal |
| Max. Range | 110 km (60 nm) maximum |
| | 092 km (50 nm) nominal |
| **Vehicle** | |
| MTOW | 7.3 kg |
| MGTOW | 9.0 kg |
| Nominal Payload Mass | 5.0 kg |
| Wingspan | 3.0 m |
| Fuselage Length | 187 cm (excl. air intake nozzle for payload) |
| Propulsion | Electrical, propeller |
| **SwiftPilot™ Flight Management** | |
| Telemetry Update Rate | 10 Hz |
| Data & Control Telemetry | 900 MHz real-time radio |
| Data Storage | SD card |
| **Payload** | |
| Nose Cone | 20.3 cm diameter, 63.2 cm length |
| Payload Available Power | 50 W |
| Payload Used Power | 1.3 W |
| Payload Mass Capacity | 3.5 kg |
| Geotagging Accuracy | <1 m (all directions) |
| Downlink Data Rate | 115200 bps (serial) |



**Appendix D: Euclidean Distance and the Calinski-Harabasz Criterion Index**

To compare clusters, a distance needs to be established. The abstract length of a vector in a real vector space is the $L^p$-norm (Eqn. 1), defined as distance $d_p$ between two points $a$ and $b$ with $m$ features where $p$ is any real number and $p>=0$.

$$d_p[a, b] = (|b_1 - a_1|^p + |b_2 - a_2|^p + \cdots + |b_m - a_m|^p)^{1/p} \tag{2}$$

The Euclidean distance ($L^2$-norm) between specific humidity and potential temperature was chosen to be an effective distance (*Toledo et al.*, 2014). In a self-similarity plot of pairwise distance between all points, the maximal distance between

points, represented as the first off diagonal, provides a predictive tool for PBL height (Fig. 7). In the clustering analysis, environmental measurements were averaged into 5 m vertical bins and normalized between 0 and 1.

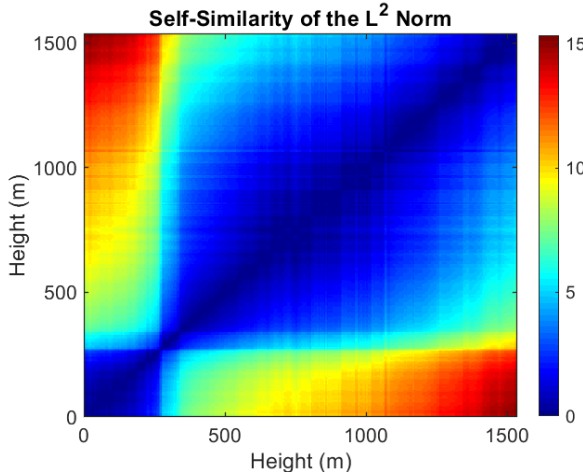

**Figure D1: Plot of the self-similarity matrix of the $L^2$-norm of atmospheric values determined for the sampling flight June 12th,**
**2019. The first off diagonal represents the comparison of every value against every other value. This method was used to identify the PBL during all 2019 flight operations, later updated with the Calinski-Harabasz criterion index.**

While Euclidean distance is more robust than individual gradient analysis (*Krawiec-Thayer* 2018), the technique still returns multiple candidates for the PBL height. Instead, indexing methods can provide a deterministic global maximum of centroid partitions associated with the dataset. For the Calinski-Harabasz criterion index, the centroid is determined with a

nonhierarchical $k$-means method. $k$-means is a data-partitioning algorithm that determines groupings of $k$ amount of centroid clusters of n total observations converging to a maximum criterion value or index between centroids. $k$ is determined *a priori* to be 2 corresponding to the assumed present atmospheric regions, the PBL and free troposphere. The Calinski-Harabasz criterion index has been used successfully with $k$-means methods in previous remote sensing and weather balloon studies (*Toledo et al.* 2014, *Caicedo et al.* 2017).






The Calinski-Harabasz index is the ratio of variance within one centroid and the variance between origin locations of all other centroids. Let $m_i$ as the centroid of cluster $i$ containing $n_i$ data points, and $c$ be an origin point for the data set. The variance within one cluster is defined below in Equation 3:

$$D_W = \sum_{x \epsilon \alpha} (d_2[x, m_a])^2 + \sum_{x \epsilon \beta} (d_2[x, m_\beta])^2 \tag{3}$$

The expression for variance between clusters is defined as

$$D_B = n_\alpha (d_2[m_a, c])^2 + n_\beta (d_2[m_\beta, c])^2 \tag{4}$$

The ratio of variances, the Calinski-Harabasz index, then follows as

$$CH = (n_\alpha + n_\beta - 2) D_B D_W^{-1} \tag{5}$$

The centroid pair with the highest index is then the most significant group of partitions and the height that corresponds with

the boundary of the two groups is assumed to be the upper layer of the PBL. An example of this method is shown in Fig. 6 and 10.



**Data Availability**

A data package upload has been initiated with Arctic Data Center, which is committed to providing citable datasets to facilitate reproducible science. Each DOI issued by the Arctic Data Center is intended to represent a unique, immutable
version of a data package. We will finalize the data package during the review process with AMT, based in part upon referee feedback for the manuscript.

**Data Availability**

The authors are grateful for the funding provided by the National Science Foundation Award 1833165, "Closing the Water Vapor Exchange Budget Between the Ice Sheets and Free Atmosphere", managed by Jennifer Mercer.  We wish to thank
Dorthe Dahl-Jensen, University of Copenhagen and the EastGRIP international team for their support of the fieldwork on the Greenland ice sheet; EastGRIP is directed and organized by the Centre for Ice and Climate at the Niels Bohr Institute, University of Copenhagen. It is supported by funding agencies and institutions in Denmark (A. P. Møller Foundation, University of Copenhagen), USA (US National Science Foundation, Office of Polar Programs), Germany (Alfred Wegener Institute, Helmholtz Centre for Polar and Marine Research), Japan (National Institute of Polar Research and Arctic
Challenge for Sustainability), Norway (University of Bergen and Bergen Research Foundation), Switzerland (Swiss National Science Foundation), France (French Polar Institute Paul-Emile Victor, Institute for Geosciences and Environmental research) and China (Chinese Academy of Sciences and Beijing Normal University); Water isotope observations above the ice sheet was supported by the European Research Council (ERC) under the European Union's Horizon 2020 research and innovation program: Starting Grant SNOWISO (grant agreement 759526); the 109[th] Air National Guard for logistical
support in reaching the remote EGRIP ice core camp, the shipment of equipment, and for safe passage for our team members; UAV expertise and design from Jack Elston and the team at Black Swift Technologies; and nose cone design assistance from Dirk Richter, University of Colorado.

**Author Contributions**
HCSL, BHV, and TRJ developed the initial idea, rationale, and experimental setup. KSR, TRJ, and BHV prepared the original draft and all authors contributed to the review and editing of this paper. Design of the UAV sample pod was a product of BHV, KSR and JE. Flights and field water isotope analysis were done by KSR, VM, BHV, TRJ, WS, and AH. Boundary layer prediction algorithms were developed by TRJ and KSR. Figures were prepared by KSR, BHV, and TRJ. For comparison to UAV flask samples, concurrent water isotope tower measurements were provided by HCSL, SW, and AKF.
Insights for modeling and error analysis were provided by HCSL, SW, and AKF.



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
