# Peer review of "An Unmanned Aerial Vehicle Sampling Platform for Atmospheric Water Vapor Isotopes in Polar Environments"

_Atmospheric Measurement Techniques, 2021_

## Author Comment (AC1)

**Reviewer 1:**

The paper contributes valuable to the ongoing evolution of using UAV/UAS in environmental science. Some minor to major revisions are suggested:

2. Introduction: The study is well motivated and hypothesized. The implications of in-situ vs. extractive measurements should be discussed.

   There is no extraction step in our sampling. We transport whole air samples to the surface via our sampling system for input to an optical instrument.

3. Line 113: The mentioned normalization to Standard Light Antarctic Precipitation is not explained

   The language has been changed to appropriately describe the isotopic tie to Standard Light Antarctic Precipitation in Section 2.1. The IAEA Certification report No. 63 states: "For the elements hydrogen and oxygen, the two international measurement standards used to calibrate all relative stable isotope ratio measurements are named VSMOW2 (Vienna Standard Mean Ocean Water 2) and SLAP2 (Standard Light Antarctic Precipitation 2) [2, 3]. The isotopic ratios of these two materials span almost the total range of isotopic compositions of natural water samples on earth. All stable isotope ratio measurements for hydrogen and oxygen performed worldwide are thus directly or indirectly calibrated versus these two international measurement standards, which have replaced the previously available water reference materials VSMOW and SLAP in the year 2006."

   Based on the above report, we have revised the text: "The data consist of measurements of hydrogen and oxygen isotopes in water vapor, where the ratio of heavy to light water isotopes in a sample is expressed in δ notation (Epstein et al. 1953, Mook 2000) relative to internationally recognized primary reference materials Vienna Standard Mean Ocean Water (VSMOW) and normalized to Standard Light Antarctic Precipitation (SLAP) in accordance with IAEA reference material (2017):

   $$\delta_{sample} = \left[ \left( \frac{R_{sample}}{R_{VSMOW}} \right) - 1 \right] * 1000$$

   where R is the isotopic ratio 18O/16O or D/H (i.e., 2H/1H). The δD and δ18O symbols refer to fractional deviations from VSMOW, normally expressed in parts per thousand (per mille or ‰). In practice, we maintain a suite of secondary reference waters that are rigorously calibrated to the primary reference materials (VSMOW and SLAP). Storage of our secondary reference waters is in accordance with methods described in IAEA Technical Note No, 43, (Newman et al. 2009)."

   REFERENCE: INTERNATIONAL ATOMIC ENERGY AGENCY, Certification Report on Value Assignment for the d2H and d18O Stable Isotopic Composition in the Water Reference Material GRESP (Greenland Summit Precipitation), Analytical Quality in Nuclear Applications Series No. 63, IAEA, Vienna (2021)."

4. If fig. 1 is a general presentation of the EastGRIP project then please at the reference.

   This is an original figure to this publication, but yes, it derives inspiration from another figure from H. C. Steen-Larsen, a co-author on this paper. Figure 1b is a map obtained from Wolfram Research and a citation is now given in the caption.

5. Line 153 to 157: This is a typical but not helpful assessment of an airborne system, as the mentioned problems can easily be solved without substantial additional cost.

   Choosing the right platform is a complex process taking many aspects into account, often non-scientific reasons like funding and access to knowledge. Addressing the relevant aspects is necessary to establish better rational based approaches rather then today often seen intuitive choices.

   Finding a platform that satisfies constraints imposed by ease of use, range of flight, payload capacity, adaptability for user fabricated modifications and budget can be challenging. We found the S2 platform met our needs and had the advantage of being locally produced.

6. The chosen platform should be described with all components, normal procedures and limitations as the paper title focusses on the technical part of the overall system. Please clear some inconsistencies in numbers (text vs. app. C vs. Black Swift Technologies homepage, e.g. payload mass.).

   We do not identify any discrepancies between the text and app. C, however it appears that there are differences between the designs given to us by manufacturer and those on the website. Ours are the current numbers. We have added important payload and platform specifications within the text at line 164 originally found in app. C.

   Revised text: "The aircraft can carry up to a 3.5 kg payload for up to 90 mins. At arctic temperatures with the payload used in this study, we found 45 mins of flight time typical and apt for climbing 1600m and including needed sampling time."

7. 2.4.1: Can you please add a system diagram to fig. 3?

   This is now included.

8. 2.4.1: Was the payload leak tested in low temperature conditions and mechanical vibrations (inflight conditions)?

   Testing was performed in an arctic freezer prior to the field campaign to confirm material decisions. The intercomparison with the separate system described in Section 2.3 was in part motivated to address concerns of the system performance in conditions on-site. Vibration was not independently tested but motivated taking samples in pairs.

   Revised text at line 364: "Paired sampling was motivated primarily by the inability to test the low temperatures, the 12G forces exerted on the flasks during launch, and inflight vibration forces in a "benchtop" setting."

9. 2.4.1: Did the choice of components and materials take into account a potential corruption of the air samples?

   We explored the use of Teflon, Tedlar, and stainless steel bags. We observed memory effects in all three of those options. We were initially apprehensive to use glass due to the potential percussive forces a flask could experience during takeoff and landing. This proved to not be true after test flights.

Revised text at line 181: "We explored and tested the efficacy of holding water vapor within Teflon, Tedlar, and stainless-steel bags and we observed memory effects in all three of those options. Glass was the only material found where sample carry over was minimal."

10. Line 186: I don´t understand the half sentence "yielding appr. 50 flasks …". I guess a time reference is missing?

   This has been addressed by adding "over the 5-minute flush-fill process for each sample" to line 193.

11. Line 199ff: I guess the temperature and humidity of the undisturbed air at the position of the UAV is meant? Please add a description of the sensor installation, as this is essential for a later discussion.

   The temperature and humidity sensor is attached approximately halfway down the right wing and is part of the combined sensor package used by the autopilot. It is installed pointing forward. Text to mention this has been added to line 209.

   Revised text: "Both sensors are included as part of the forward pointing package to assist in autopilot flight on the right wing of the aircraft."

12. Line 207ff: "Flasks … opening a single port on the flask." Does this mean that the Picarro-System sucks air out of the flasks, which reduces the pressure inside? If yes, is there an influence on the isotopic result because of condensation?

   There is an influence because of condensation of the water vapor, but only at the point when there is a strong pressure gradient between the flask and the cavity. For the initial sampling period of a few minutes, the measured isotope value is effectively constant. We will report the stability of the isotope value over the course of the integrated measurement period in the data product. We have added text to address this in line 219.

   Revised text: "In this manner the Picarro analyzer is pulling the sample air from the dead end of the flask, reducing the pressure slowly over time."… "Additionally, to address any issues associated with any reductions in flask pressure near the end, the last 3 minutes are also cropped."

13. Line 209ff: I guess you have experiences in appropriate flushing and filling times. Can you please explain this a bit more detailed or cite a proper reference?

   We have added the appropriate explanation to line 225

   Revised text: "These timings were empirically derived from consistent plateaus of both isotopes and water concentration between the beginning and ending tails."

14. 2.5: Are pressure differences between flasks and Picarro system an issue?

   The Picarro instrument maintains a controlled pressure within its measurement cavity. Text has been added to line 221 to mention this.

   Revised text: "Pressure within the analyzer cavity is carefully controlled at 50 Torr by the instrument with high speed PID controlled valves on both ends of the cavity."

15. 2.6: The trials described in this section make the most of the possible, which is much more then often seen. I suggest discussing the difference between airborne and stationary samplings (e.g. fluxes?). Standard deviation and root mean square lead to the same result, but have other constrains. So I suggest to be careful presenting 1 sigma values.

We apologize, but we aren't exactly sure what is meant by this comment. As for the difference between airborne and stationary samples, stationary samples were used to establish an empirical uncertainty by sampling all flasks from all three pods over the course of 90 mins. Built into that is an implicit assumption that we are sampling the "same air" over that time period. This may not be true and sampling is subject to diurnal variation at that time scale. This is why we report our uncertainty as the worst-case scenario as water vapor from standard water would likely outperform outside air.

16. Make sure, that sec 2.4 and 2.8 do not double each other.

In section 2.4, we clarify that a typical analysis day will be explained in 2.8, to alert the reader that additional information will be forthcoming in a later section. Section 2.4 lays out each individual process with appropriate technical details. Section 2.8 lists those methods and shows how those methods are used during a sampling day. With that said, we feel that each section is worth keeping to ensure clarity on measurement scheme vs. typical day.

Revised text: "A typical flight day including sampling is found in Section 2.8."

17. Line 347: As altitude maintaining power setting you normally do not use "idle".

Language has moved from "idling" to "holding altitude".

18. Line 376: Time reference (e.g. UTC)?

Fixed.

19. Fig 7.: The overlaid periodic changes in temperature and humidity correspond to the heading of the UAV. As the installation is not described (comment X.) nor it is clear if the airspeed or the ground speed is commanded by the autopilot it cannot be excluded that an improper installation and/or data correction leads to this result. At this point I do not agree with the options described in lin 434ff as better sensors cannot heal improper installations.

The airspeed and the ground speed are not determined by the pilot. The autopilot solves for the desired path and climb/descent rate. Conceptually, there exists a tradeoff between controlling climb rate and air speed but that decision is outside the purview of the pilot during flight. Wind speed is measured and used as part of the autopilot but not of appropriate reportable quality for making a correction in this study. Since submitting this document for review, we have obtained the ability for our system to determine reportable 2D wind speeds and we have included text within Conclusions and Outlook at line 480 to mention the new sensor.

Revised text: "We plan to leverage an existing anemometer used by the autopilot in order to assist in the correction as well as produce an additional 2D wind speed for the flight."

20. Line 582: Acknowledgements

Fixed.

Additional comment:

In the time since submission, we have determined that the first two flights of our six total do not have sufficiently useful calibrations. This was discovered from referencing lab notebooks from which it was found that calibration protocols were not correctly followed at the beginning of the field campaign. The midair isotope values for both the June 3rd and June 6th flights are precise and the midair isotope gradients real, but the values are not accurate to the standard necessary for reporting in this paper. To remain conservative in this pilot study, we are choosing to omit the data from the text and data product. Please note, this does not change any conclusions in the paper and only necessitates minor explanatory changes in the discussion, which has already focused on the June 12th flight onwards.

---

## Author Comment (AC2)

**Reviewer 2:**

This paper describes the use of a professional UAV for atmospheric sampling, in this example for water vapour isotopes above central Greenland. Such a platform is a welcome addition to ground-based measurements, especially under hars conditions such as in this paper.

The data presented are still only a few, not enough for new scientific insights, but enough for the proof of principle, and as such this paper fits the AMT journal.

I have several (smaller) comments and remarks, and I invite the authors to use them for slight changes in the paper.

2.4  I'd like a bit more details on the used UAV: power, total stored energy, the actual payload in kgs, and the way of launching (only later in the paper it becomes apparent it is launched by some 12G launching system). Of course, the list is in the appendix, but some key numbers in the main text would be welcome.

We have added important descriptors for the platform to the text in Section 2.4, line 161.

Revised text: "The S2 is capable of conducting fully autonomous flights in unimproved areas such as an ice sheet in part due to its pneumatic launch system. The aircraft can adjust to changing wind conditions in real-time, ensuring a high degree of stability for predefined mapping or atmospheric sampling applications (Elston et al. 2015b). The aircraft can carry up to a 3.5 kg payload for up to 90 mins. At arctic temperatures with the payload used in this study, we found 45 mins of flight time typical and apt for climbing 1600m and including needed sampling time."

What is the mass of the flasks used?

We have added the mass of the flasks (181 grams for each flask) and volume is ~ 500 cc on line 188 of manuscript.

Revised text: "The glass flasks are approximately 181 grams each, 500cc in volume, and include a supported dip tube to ensure the sample is adequately flushed during fill."

Table 1: the numbers within parentheses seem too large, especially for SPGSW, which is so close to SLAP2 that virtually only that uncertainty would add up to the primary uncertainty: $(0.2^2+0.3^)\approx0.36$‰

We do not calibrate our secondary references to the closest primary standard. Instead, we calibrate to an interpolated line of the correction amount determined from all the primary standards. As such, we include all primary standards in the error propagation. For SPGSW that would be $(1.2^2+0.3^2+0.3^2+0.2^2)^{(1/2)} = $ ~1.288

We have included a reference for this procedure to the caption of Table 1.

Revised text: "Additional details describing the calibration scheme can be found in *Jones et al.*, 2017."

line 290 the ‰ sign (or the word per mil) is lacking twice

Fixed.

Figure 5 and text. Sure d-excess is a powerful comparison; nevertheless, the individual isotopes D and 18O themselves would be equally interesting. As d-excess from the pods is somewhat higher than that of the tower, it would be interesting to see if this is caused by D or 18O, or an interplay of both.

In an earlier draft of the paper, we did show both but the graphs were so identical, with both D and 18O and their relative difference from the tower setup, that we decided just to report dxs. We believe it is an interplay

of both. A line to mention this has been added to line 310. As well, too additional figures have been added to Appendix A separating the two different isotopes.

Revised text: "The positive relation is seen in both δD and δ¹⁸O implying that the positive bias is due to an interplay of both measurements. Figures of separated δD and δ¹⁸O can be found in Appendix A."

Line 308: The "Euclidean distance in the measurement domain" is also explained in appandix D, please mention that in the text.

We have added a second reference to appendix D.

And what is the difference between that and 'just' taking  for example the average of the height of the max gradient in water vapor content and similar in other parameters?

In a way, we are already doing that though measuring water vapor content is something we can measure for flasks on the ground and not airborne. Euclidean distance between observation points is a gradient, just the normalized aggregate of all observable parameters. The reason we use the clustering index is because gradients alone returns multiple candidates for PBL location where the index considers the difference in 'likeness' between two regions and is less ambiguous. Discussion for this difference is in Appendix D.

Figure 6 also indicate the PBL that you actually took, based on your in-flight method at the time.

Added.

lines 345-350. While I of course see the advantage of taking duplicate samples, the alternative would have been 6 (or 8) altitudes, with also advantages! Why has the duplicate choice been made? (or one duplicate and 4 single ones, or any other combination). Would you choose differently for the next campaigns? May be something to discuss in the conclusions and outlook chapter?

This was certainly a difficult decision! We decided on pairs because of the difficulty of creating a benchtop equivalent test of both the low temperatures and mid-flight vibration exerted on the flasks in the field. We believe that, after the apparent reproducibility of pairs from this field campaign, pairs won't be needed. As well, by alternating sample pods used on the drone vs taking the 2 meter sample, we can determine if any of the pods are experiencing leaks leading to a previous flight with such a pod being flagged for bad data. We have added a line to mention this.

Revised text at line 364: "Paired sampling was motivated primarily by the inability to test the low temperatures, the 12G forces exerted on the flasks during launch, and inflight vibration forces in a "benchtop" setting."

Line 373 what is AGL ? Caption figure 7 gives meters ABL  ? The plots all simply state "height"

The typo of ABL has been changed to AGL in Figure 7. AGL is "above ground level" and the acronym is defined at line 71. We also define it now in the Figure 7 caption.

Fig 8 only shows the d2H (18O is in the appendix). Apparently the authors do not think the isotope measurements are worth a 2H-18O relation plot (or d-excess for that matter)? Perhaps the July 12 points would be worth a plot or table for the 2H-18O relation?

Or alternatively still a third horizontal axis in fig 8 and show the 18O's as well (different marks color, and slightly displaced in height).

We choose not to originally include a 2H-18O relation plot as we intend for this paper to focus on the process of producing the isotope values for water vapor in air. That said, we have included it as an appendix figure and referenced it in the caption of Figure 8 where the 18O data is referenced as well. We have also now included dxs as a column of the data product.

The conclusions are a bit long compared to the rest of the text, so it can be shortened, and renamed "conclusions and outlook", as the last paragraph is about the future perspective (and need not be shortened).

We have significantly reduced the length of the conclusion without affecting the future perspective portion of it.

Revised text: "We have presented a UAV-isotope sampling platform and methodology capable of measuring atmospheric water vapor and its stable isotopes within the planetary boundary layer (PBL) and lower troposphere in a polar environment. We utilize a fixed-wing UAV (Black Swift Technologies) with flight times in excess of 45 minutes with the capability to reach 1,600m AGL. Multiple nose cones allow for collection of air in 8 glass flasks, enclosed within a 3D printed support structure that critically withstands 12Gs of force during takeoff. In this study, the total system is used to sample above and below an algorithmically-detected PBL, resulting in the first measurements of atmospheric water isotopes above and below the PBL on the high-altitude Greenland Ice Sheet.

Across four sample-taking missions at the EGRIP ice core site in 2019, we observed significant variation in water isotopes on either side of the PBL; the variability exceeded our conservative precision estimates of 2.8‰ in $\delta D$ and 0.45‰ in $\delta 18O$. These results form the basis for future campaigns to collect high-temporal density measurements (flights every 4-6 hours across many weeks) at key missing scales that will improve ice-to-atmosphere modeling and mixing processes, ice sheet mass balance, satellite detection algorithms, moisture tracking, ice core science, and modeling the hydrologic cycle in general.

A field campaign for return to EastGRIP is scheduled for summer 2022. Future improvements to the UAV-isotope system will be primarily focused on logistical improvements that increase the number and frequency of flights. Additional flight crew will be available for nighttime flight missions. To ensure a balanced diurnal flight schedule over weeks of time, with the goal of one flight every 4-6 hours, a precessing schedule of calibration times will be used. Each calibration will be done every 2-4 days, lasting 12 hours, starting at different times of day. This ensures that we do not consistently lose the ability for UAV sampling at the same time for every calibration, e.g. from 12pm-12am. The combination of these improvements will allow the potential maximum number of flights per day to increase from two to as many as six, while balancing the timing of calibration. In flight, we will carefully regulate the rate of ascent and include better performing temperature and humidity sensors with minimal time constants, all of which will reduce hysteresis for PBL detection. We plan to leverage an existing anemometer used by the autopilot in order to assist in the correction as well as produce an additional 2D wind speed for the flight. Additional improvements will include a lighter pump and manifold system that should allow greater flight time. Beyond Greenland, this platform is readily adaptable to other scientific disciplines, and will be used in an upcoming permafrost project to measure atmospheric methane emissions and soil moisture content in Alaska."

Additional comment:

In the time since submission, we have determined that the first two flights of our six total do not have sufficiently useful calibrations. This was discovered from referencing lab notebooks from which it was found that calibration protocols were not correctly followed at the beginning of the field campaign. The midair isotope values for both the June 3rd and June 6th flights are precise and the midair isotope gradients real, but the values are not accurate to the standard necessary for reporting in this paper. To remain conservative in this pilot study, we are choosing to omit the data from the text and data product. Please note, this does not change any conclusions in the paper and only necessitates minor explanatory changes in the discussion, which has already focused on the June 12[th] flight onwards.

---

## Author Response (AR2)

**To the editor**: We appreciate the reviewer's comments on all fronts. To address the reviewers request for a description of the installation of the temperature sensor, we have added the full part number for the sensor which details the package as well as a clarification to where the sensor package is mounted in section 2.4.2. We also include the data sheet as a citation for readers curious about the mechanical design of the sensor package. While we don't believe that top-down solar radiation significantly affects our temperature measurement, we neglected to consider the upwards-pointing surface albedo effect over the ice sheet. We now mention this in the text. To address to impact of angle of attack of our aircraft on sensor hysteresis, we tied our existing discussion about reducing bank angle to new discussion about the impact of the angle of attack. We include mention of our plan to test ascension rates in our future field campaigns to reduce the hysteresis effect. Changes to the text are shown by yellow highlights below:

**New text in first paragraph in 2.4.2**

[revised manuscript text omitted]

---

## Author Response (AR3)

In this minor revision, we addressed:
- Revised formatting for references.
- Added text for 'Competing Interests'.
- Provided a full-text Table in Appendix C.

Please note. We are still awaiting a DOI from the Arctic Data Center. We expect to have the DOI within two weeks.